# Hepatic Steatosis Can Be Partly Generated by the Gut Microbiota–Mitochondria Axis via 2-Oleoyl Glycerol and Reversed by a Combination of Soy Protein, Chia Oil, Curcumin and Nopal

**DOI:** 10.3390/nu16162594

**Published:** 2024-08-06

**Authors:** Mónica Sánchez-Tapia, Sandra Tobón-Cornejo, Lilia G. Noriega, Natalia Vázquez-Manjarrez, Diana Coutiño-Hernández, Omar Granados-Portillo, Berenice M. Román-Calleja, Astrid Ruíz-Margáin, Ricardo U. Macías-Rodríguez, Armando R. Tovar, Nimbe Torres

**Affiliations:** 1Departamento de Fisiología de la Nutrición, Instituto Nacional de Ciencias Médicas y Nutrición Salvador Zubirán, México City 14080, México; monica.sanchezt@incmnsz.mx (M.S.-T.); sandra.tobonc@incmnsz.mx (S.T.-C.); lilia.noriegal@incmnsz.mx (L.G.N.); natalia.vazquezm@incmnsz.mx (N.V.-M.); dianacristalch@gmail.com (D.C.-H.); omar.granadosp@incmnsz.mx (O.G.-P.); armando.tovarp@incmnsz.mx (A.R.T.); 2Departamento de Gastroenterología, Instituto Nacional de Ciencias Médicas y Nutrición Salvador Zubirán, México City 14080, México; berermn@gmail.com (B.M.R.-C.); astrid.ruizm@incmnsz.mx (A.R.-M.); ricardomaciasr@incmnsz.mx (R.U.M.-R.)

**Keywords:** hepatic steatosis, functional foods, gut microbiota, 2-oleoyl glycerol, mitochondria function, ROS

## Abstract

Metabolic dysfunction-associated steatotic liver disease (MASLD) is a serious health problem, and recent evidence indicates that gut microbiota plays a key role in its development. It is known that 2-oleoyl glycerol (2-OG) produced by the gut microbiota is associated with hepatic fibrosis, but it is not known whether this metabolite is involved in the development of hepatic steatosis. The aim of this study was to evaluate how a high-fat–sucrose diet (HFS) increases 2-OG production through gut microbiota dysbiosis and to identify whether this metabolite modifies hepatic lipogenesis and mitochondrial activity for the development of hepatic steatosis as well as whether a combination of functional foods can reverse this process. Wistar rats were fed the HFS diet for 7 months. At the end of the study, body composition, biochemical parameters, gut microbiota, protein abundance, lipogenic and antioxidant enzymes, hepatic 2-OG measurement, and mitochondrial function of the rats were evaluated. Also, the effect of the consumption of functional food with an HFS diet was assessed. In humans with MASLD, we analyzed gut microbiota and serum 2-OG. Consumption of the HFS diet in Wistar rats caused oxidative stress, hepatic steatosis, and gut microbiota dysbiosis, decreasing α-diversity and increased *Blautia producta* abundance, which increased 2-OG. This metabolite increased de novo lipogenesis through ChREBP and SREBP-1. 2-OG significantly increased mitochondrial dysfunction. The addition of functional foods to the diet modified the gut microbiota, reducing *Blautia producta* and 2-OG levels, leading to a decrease in body weight gain, body fat mass, serum glucose, insulin, cholesterol, triglycerides, fatty liver formation, and increased mitochondrial function. To use 2-OG as a biomarker, this metabolite was measured in healthy subjects or with MASLD, and it was observed that subjects with hepatic steatosis II and III had significantly higher 2-OG than healthy subjects, suggesting that the abundance of this circulating metabolite could be a predictor marker of hepatic steatosis.

## 1. Introduction

Metabolic dysfunction-associated steatotic liver disease (MASLD) refers to fatty liver disease related to systemic metabolic dysregulation [1] and adverse alterations in glucose, fatty acid, and lipoprotein metabolism with or without inflammation. In MASLD, triglycerides (TG) accumulate in the liver due to an imbalance between lipid storage and lipid oxidation. MASLD is due to (a) hepatic fatty acid uptake derived from plasma free fatty acid (FFA) released from TG hydrolysis in adipose tissue [2], (b) hepatic de novo fatty acid synthesis regulated independently by insulin and glucose through the activation of SREBP-1c and ChREBP, respectively, and (c) transport of fatty acids into the mitochondria for oxidation, which is regulated by carnitine palmitoyl transferase (CPT1), carnitine translocase, and CPT2. These enzymes are regulated by the transcription factor PPARα; however, it is not clear if mitochondrial oxidative enzymes lead to hepatic steatosis since normal, increase or decrease of fatty acid oxidation in subjects with hepatic steatosis has been demonstrated [3]. The loss of mitochondria function is implicated in metabolic diseases such as diabetes and obesity, and the development of new therapies aimed at improving mitochondrial function has become a primary focus of attention [4]. Another important factor to consider is the role of the gut microbiota in the development of hepatic steatosis. The liver represents the first point of contact for bacteria and microbial components through bile and enterohepatic circulation [5]. There is evidence that intestinal microbiota is involved in the regulation of lipid metabolism, including cholesterol and triglycerides. It has been suggested that the gut microbiota may regulate hepatic lipid metabolism through the generation of metabolites, including short-chain fatty acids or bile acids [6,7]. However, evidence that a specific metabolite regulates hepatic lipogenesis or fatty acid oxidation, leading to changes in hepatic lipid accumulation, as observed in MASLD, remains limited. As MASLD has become a major disease in Western society, it is important to determine how the gut microbiota–liver axis may trigger lipid accumulation in the liver. Hence, potential metabolites implicated in hepatic steatosis could be modified by consuming functional foods rich in bioactive compounds such as soy protein [8,9], chia oil [10], curcumin [11], and nopal [12], which are known to modify the taxonomy of the gut microbiota and this could reduce hepatic lipid accumulation. Therefore, the aim of the present work was to study the effect of the consumption of a high-fat–sucrose (HFS) diet on the taxonomy of the gut microbiota, in particular looking for species that could be associated with the presence of hepatic steatosis and to determine whether the production of a specific metabolite could be a link between the gut microbiota and fatty liver through modification in the abundance of enzymes involved in lipogenesis and fatty acid oxidation, as well as in mitochondrial function. Furthermore, to establish whether the consumption of a combination of functional foods rich in bioactive compounds could reverse changes in the gut microbiota, which could prevent the activation mechanism leading to the development of hepatic steatosis.

## 2. Materials and Methods

### 2.1. Animal Study Design

Male Wistar rats aged 5–7 weeks were acquired from the National Institute of Medical Sciences and Nutrition. The animals were housed in individual cages, with access to food and water ad libitum (12 h/12 h light–dark cycle, 22 °C, and 40–50% relative humidity). The protocol of this study was approved by the Bioethics Committee of the Instituto Nacional de Ciencias Médicas y Nutrición Salvador Zubirán, Mexico City (FNU1444). To calculate the sample size, the formula for comparison of means was used [13]. Rats were randomized into two groups: (i) a control group (C, n = 6) fed a control diet according to the American Institute of Nutrition recommendations (AIN-93) [14], and (ii) a high-fat–sucrose diet (HF + 5%S) consisted of a high-fat diet and 5% sucrose added to the drinking water (HFS, n = 12). In the fourth month, biochemical and physiological parameters were measured in plasma, including glucose, triglycerides, and cholesterol levels, glucose tolerance test, body composition, and energy expenditure. Intervention phase: HF + 5%S group was divided into 2 groups: HF + 5%S diet (HFS, n = 6), and the group fed functional foods HF + 5%S + FF (HFS + 5% + FF, n = 6), for three months. The C group continued consuming the AIN-93 diet (C, n = 6), (Table 1. Diet composition). Once a week, bottles were replaced with fresh solutions. Animal weight and food consumption were recorded every other day during the protocol. At the end of this period, liver, serum and feces samples were immediately taken and stored at −70 °C.

### 2.2. Human Study

Thirty-four subjects were studied: 24 healthy subjects and 10 subjects with MASLD between 18 and 60 years old. Subjects with MASLD were attending the liver unit at the National Institute of Medical Sciences and Nutrition Salvador Zubirán. The study followed the principles according to the Declaration of Helsinki and was approved by the Research Ethics Committee of the Institution (Ref. 3608). The diagnosis of liver steatosis was confirmed by elastography using UVCTE. BMI, biochemical variables, and elastography were evaluated. We determined serum 2-OG, gut microbiota, and in this case–control study to investigate the link between gut microbiota dysbiosis, 2-OG and MASLD.

### 2.3. Body Composition Analysis

Body composition was measured by a quantitative magnetic resonance imaging system (Echo MRI, Houston, TX, USA). Rats were placed into a plastic cylinder with a cylindrical plastic insert added to limit movement. While in the tube, animals were briefly subjected to a low-intensity (0.05 Tesla) electromagnetic field to measure body fat and lean mass. Measurements were conducted before and after the dietary treatments.

### 2.4. Biochemical Parameters

Serum glucose, total cholesterol, and triglycerides were determined by colorimetric enzymatic assay with the COBAS C111 Analyzer (Roche, Basel, Switzerland). Serum lipopolysaccharide and insulin were determined according to the instructions of commercial ELISA kits (Cloud-Clone Corp, Houston, TX, USA, cat. CEB526Ge and Alpco Diagnostics, cat. 80-INSRT-E10 Salem, NH, USA).

### 2.5. Histological Analysis

Liver samples were fixed in 10% phosphate-buffered formalin and embedded in melted paraffin blocks. After the paraffin solidified, the blocks were cut to a thickness of 4 µ, which were stained with 95% hematoxylin-eosin. The histological sections were observed under a Leica microscope stained with hematoxylin and eosin (Leica DM750 Wetzlar, Germany).

### 2.6. Western Blot Analysis

Liver samples were collected after 8 h of fasting. Total liver protein was extracted with a lysis buffer (RIPA) and quantified by Bradford assay (Bio-Rad, Hercules, CA, USA) and stored at −70 °C. Protein samples from liver or hepatocytes were separated on acrylamide gels and blotted onto PVDF membranes. Membranes were blocked in a blocking buffer consisting of 2% BSA in TBS tween or 5% fat-free milk (hepatocytes) and incubated overnight at 4 °C with the primary antibody. The blots were incubated with secondary antibodies conjugated with horseradish peroxidase. GAPDH was used to normalize the data. Images were analyzed with a ChemiDocTM XRS + System Image LabTM Software 6.1 (Bio-Rad, Hercules, CA, USA). The membranes were incubated with specific antibodies against FAS (sc-20140 1:1000), PPARα (sc-398394 1:1000), and ChREBP (sc-21189 1:10,000). ACC (MA5-15025 1:1000), AGPAT6 (PAS-49623 1:2500), CPT1 (ab128568 1:2000), GAPDH (ab201822 1: 20,000), SREBP1 (ab3259 1:1000). The proteins of interest were detected with goat anti-mouse IgG-H&L antibody (ab6789 1:20,000), or goat anti-rabbit IgG-H&L (ab6721 1:20,000).

### 2.7. 16S rRNA Sequencing

Fresh fecal samples were collected at the end of the study and were immediately frozen and stored at −70 °C until use. Bacterial DNA content was extracted using the QIAamp DNA Mini Kit (Qiagen, Valencia, CA, USA) according to the manufacturer’s instructions. DNA samples were amplified using V3 and V4 regions and overhang adapters attached. After a capillary electrophoresis, the size of the fragment was checked and purified. The Nextera XT index kit was used to add indices and Illumina sequencing adapters. After the validation of the library, the samples were pooled and injected into the MiSeq system according to the manufacturer’s instructions. At the end of the run, the MiSeq system generates FASTQ files for downstream analysis.

### 2.8. Primary Hepatocyte Cell Culture and Mitochondrial Function

Rat primary hepatocytes were collected by in situ perfusion of the liver according to the method by Berry and Friend [15]. The procedure consisted of cannulating and exsanguinating the liver in vivo, followed by a continuous perfusion with collagenase. Then, the liver was isolated and placed in Hanks balanced salt solution (HBSS) to disaggregate the tissue. The cell suspension was filtered through a 100 μM mesh, washed twice with HBSS, and re-suspended in DMEM-F12 medium without HEPES supplemented with 10% FBS and 1X antibiotic. Rat primary hepatocytes were seeded into an XFe96 microplate at a density of 10,000 cells/well. The medium was changed after 4 h to remove unattached cells. Cells were then incubated with 0, 1, 3, and 5 μM of gallic acid equivalents of an ethanolic extract of the experimental diet for 18 h. Mitochondrial function was then evaluated by performing a mitochondrial stress test in an extracellular flux analyzer XFe96 (Agilent Technologies, Santa Clara, CA, USA). Briefly, cells were washed and incubated for 1 h in a non-CO_2_ incubator with an XF basal medium supplemented with 10 mM glucose, 1 mM pyruvate, and 2 mM glutamine. For the experiment, 2 μM oligomycin, 0.5 μM FCCP, 1 μM rotenone/antimycin A, and 50 mM of 2-deoxyglucose were sequentially injected, and three measurements were performed in basal conditions and after the addition of each compound. Basal mitochondrial respiration, ATP-linked respiration, proton leak, maximal respiration, non-mitochondrial respiration, and spare respiratory capacity were then calculated based on the OCR values. Glycolysis, glycolytic reserve, and glycolytic capacity were calculated based on the ECAR values as previously described [16].

### 2.9. Determination of 2-Oleoyl-Glycerol Analysis by MRM-IDA-EPI

For lipid extraction, 25 µL of plasma was treated with 975 µL H_2_O, 2 mL MeOH, and 0.9 mL CH_2_Cl_2_. In the case of liver, 50 mg of liver tissue was homogenized and treated with 965 mL H_2_O, 2 mL MeOH, and 3.8 mL CH_2_Cl_2_. The metabolomic analysis was performed on an Exion LC-AD linked to a QTRAP 6500+ with an Ion-Drive Turbo V source (Sciex, Framingham, MA, USA). A 2 µL sample was separated on a Luna NH_2_ HPLC column (100 × 2 mm, 3 µm, 100 Å) with an NH_2_ guard (Phenomenex, CA, USA). Chromatographic separation utilized 2 mM Ammonium Acetate in Dichloromethane/Acetonitrile (7/93) (Sigma Aldrich/JT Baker) (A) and Water/Acetonitrile (50/50) (B). Conditions: 10%B isocratic, 0.3 µL/min, 4 min. Quantification and identity confirmation used MRM, IDA, and EPI under positive ionization (5200 V, 400 °C TEM, CAD low, CUR 20 psi, Gases 1 and 2 at 45 and 50 psi, EP 10 V). MRM transitions were mz = 357.9/265.1 (DP = 46, CE = 15, CXP = 24), and 357.9/339.7 (DP = 46, CE = 13, CXP = 24 V). The calibration curve was drawn with a 2-OG standard (Cayman) in MeOH (0.015 µg–2 µg). Data analysis was conducted with SciexOS 1.7.2, estimating compound concentrations from peak areas.

### 2.10. Incubation of Hepatocytes with 2-Oleolyl Glycerol (2-OG)

Six-well plates were seeded with mice hepatocyte cells at a density of 1 × 10^6^ cells/well in 2 mL of DMEM F12 without HEPES (Gibco, Grand Island, NY, USA). The cells were stimulated with 2.5 µM of 2-OG, and after 12 h, proteins were extracted using lysis buffer (50 mM Tris (pH 7.4), 150 mM KCl, 1 mM EDTA, 1% NP-40, 5 mM NAM, 1 mM sodium butyrate, and protease inhibitors). The protein concentration of the samples was determined according to the Lowry assay (Bio-Rad).

### 2.11. Statistical Analysis

Results are expressed and plotted as mean ± SEM. Differences were considered significant when *p* < 0.05. When we compared groups before and after dietary treatments, a paired student’s *t*-test was used. When the three groups were compared at the end of the study, one way ANOVA was used. The statistical analysis was performed using Graph Path 9.0 (Graph Pad, San Diego, CA, USA)

### 2.12. Bioinformatic Analysis

The bioinformatic analysis starts with the FASTQ files that contain the sequence data from the clusters that pass the filter on a flow cell. The first step in FASTQ file generation is demultiplexing to assign clusters to a sample based on the cluster’s index sequence. The next step is clustering the pre-processed sequences into ASVs. Microbial sequence data were pooled for ASVs comparison and taxonomic abundance analysis. Alpha diversity is calculated to determine the observed richness (number of taxa) of an average sample. We quantified beta-diversity as the variability in community composition among groups and measured UniFrac as a distance metric for comparing groups and weighted and unweighted distance matrices, coupled with standard multivariate statistical techniques including principal coordinates analysis (PCoA) to identify factors explaining differences among groups using QIIME software v.2022.8. Analysis of similarities (ANOSIM) and permutational multivariate analysis of variance (ADONIS) were carried out to determine differences among groups. Differences in the relative abundance at the species levels were performed using Linear Discriminant analysis (LDA) to determine the ASVs most likely to explain differences between treatment groups with the tool LEfSe in the Galaxy platform [17].

### 2.13. Phylogenetic Investigation of Communities by Reconstruction of Unobserved States (PICRUSt)

PICRUSt2 v 2.5.0 analysis was used to predict possible metabolic pathways activated in the gut microbiota [18].

## 3. Results

### 3.1. Consumption of a High-Fat–Sucrose Diet Produced Biochemical Abnormalities Associated with the Development of Hepatic Steatosis

Rats were fed HFS or control diets for 4 months, body composition and biochemical parameters were assessed, and subsequently, the animals continued with the corresponding diet, either HFS or control diets, for an additional 3 months (Figure 1A). It was observed that after 3 months, the animals on the HFS diet gained significantly more body weight (Figure 1B) and body fat and lost significantly more lean mass than the control group (Figure 1C). Subsequently, the animals were continued on the corresponding diet, either HFS or control, for an additional 3 months. Rats consuming the HFS diet after the first 4 months showed significantly higher levels of all biochemical parameters than the C group. The difference in circulating levels of glucose (Figure 1D), insulin (Figure 1E), cholesterol (Figure 1F), and triglycerides (Figure 1G) were even greater than the control group after 7 months of HFS diet. At the end of the study, the liver of rats fed the HFS diet showed a significant increase in lipid vesicle content compared to the C group (Figure 1H), indicating the generation of a hepatic steatosis model.

### 3.2. Consumption of a HFS Diet Modified the Taxonomy of the Gut Microbiota Generating a Pro-Inflammatory State

Consumption of an HFS diet significantly reduced the α-diversity of gut microbiota compared to the C group (Figure 2A). This difference was clearly observed in the β-diversity of the gut microbiota, as it showed a significant difference in the principal component analysis between the microbiota of the animals fed the HFS diet compared to the C group (Figure 2B). The LDA analysis showed that in consumption of the HFS diet, the species that increased significantly were *Haemophilus parainfluenzae*, *Helicobacter apodemus*, and *Blautia producta* (Figure 2C). The PICRUSt analysis, which shows the bacterial metabolic pathways modified by the consumption of the HFS or C diet, showed that the consumption of the HFS diet increased the synthesis and elongation of fatty acids, as well as a decrease in the synthesis of short-chain and unsaturated fatty acids, as well as secondary bile acids (Figure 2D). It also showed an increase in the synthesis of proteins involved in the pro-inflammatory response, including lipopolysaccharide (LPS) synthesis (Figure 2E). In fact, there was a significant increase in serum concentrations of LPS (Figure 2F). This was accompanied in the colon by an increase in the protein abundance of IL-1β, TNFα, and IL-6, indicating a pro-inflammatory state, and there was a decrease in the abundance of occludins, proteins associated with tight junction stability and colonic barrier function (Figure 2G,H).

### 3.3. Synthesis of 2-Oleoyl Glycerol by Gut Microbiota Stimulates Fatty Acid Synthesis in Hepatocytes

It has been shown that the metabolite 2-OG increases in the liver of mice with hepatic fibrosis, and it was found that *Blautia producta* produces this metabolite [19]. In fact, we observed in the experimental animal study that there was a highly significant correlation between *Blautia product* abundance and 2-OG concentrations in the liver. To investigate the possible effect of 2-OG on lipid metabolism, hepatocytes were incubated with 2.5 µM 2-OG. Interestingly, we found that 2-OG significantly increased the protein abundance of the transcription factor sterol regulatory element binding protein-1 (SREBP-1), carbohydrate response element binding protein (ChREBP), acetyl-CoA carboxylase (ACC), stearoyl-CoA desaturase-1 (SCD-1) and glycerol-3-phosphate acyltransferase (AGPAT6) involved in fatty acid and triglyceride biosynthesis. We also observed an increased abundance of 3-hydroxy-3-methyl-glutaryl-coenzyme A reductase (HMGCoAr), an enzyme involved in cholesterol synthesis (Figure 3A,B). In contrast, 2-OG significantly reduced fatty acid oxidation through the transcription factor peroxisome proliferator-activated receptor alpha (PPARα) and its target enzyme carnitine palmitoyl transferase (CPT-1) (Figure 3A,B), indicating that 2-OG was able to increase lipogenesis and cholesterol synthesis and reduce fatty acid oxidation. To assess the effects of 2OG on mitochondrial function in primary rat hepatocytes, we performed a mitochondrial stress test. Notably, 2-OG significantly decreased maximal respiration and reserve respiratory capacity), without affecting basal or ATP-bound respiration, proton leakage, and non-mitochondrial respiration (Figure 3C,D). In addition, 2-OG also decreased basal glycolysis and non-glycolytic acidification (Figure 3E,F). These results suggest that 2-OG decreases the metabolic capacity of primary rat hepatocytes by altering two ATP production pathways: that of mitochondrial activity and that of glycolysis. These results indicate that 2-OG decreased fatty acid oxidation, which may lead to lipid accumulation in the liver.

### 3.4. Consumption of a HFS Diet Increases 2-OG Stimulating Fatty Acid Synthesis in the Liver Leading to Hepatic Steatosis

After understanding the mechanism of action of 2-OG, we determined the concentration of this lipid in the liver of rats fed either HFS or a control diet. The results showed a significant increase in the concentration of 2-OG in animals fed the HFS diet (Figure 4A). Interestingly, it was found that in these animals, there was an increase in the abundance of the transcription factor ChREBP and its target genes FAS, SCD1, and GPAT, as well as a decrease in the phosphorylation of pACC, indicating, as observed in hepatocytes, that increased 2-OG stimulates lipogenesis (Figure 4B,C). In fact, macroscopically, the liver of animals fed HFS diet showed a significant accumulation of hepatic lipids (Figure 4D) and liver triglycerides and cholesterol concentration (Figure 4E). Notably, there were differences in the concentration of the type of fatty acids accumulated in the liver between animals fed the HFS diet compared to control, indicating in part the influence of dietary lipid composition (Figure 4F). In summary, there was a significant correlation between *B. producta* and 2-OG, liver triglycerides, and FAS, and 2-OG with CHREBP and SCD-1, indicating the importance of the metabolite 2-OG in lipogenesis (Figure 4G).

### 3.5. A Diet Based on Functional Foods Modifies the Gut Microbiota Reducing 2-OG and Hepatic Lipid Accumulation

The evidence obtained in the study suggested that preventing a change in the taxonomy of the gut microbiota, particularly Blautia producta, despite the consumption of an HFS diet, would decrease 2-OG, which could impact a lower accumulation of fat in the liver. For this purpose, rats were fed an HFS diet for 4 months and then fed the HFS diet for 3 months with the addition of a combination of functional foods containing vegetable protein, chia oil rich in omega-3 fatty acids, dehydrated nopal rich in flavonoids, soluble and insoluble fiber, and curcumin, which are rich in fiber, bioactive compounds and have high antioxidant capacity, and the metabolic response was compared with animals fed only the HFS diet for 7 months (Figure 5A). The group fed the HFS diet had a 42% greater weight gain than the HFS +FF group, and these animals showed less body fat mass and more lean mass (Figure 5B,C). These findings were also reflected in a significant decrease in serum glucose (39%), insulin (30%), cholesterol (24.9%), and triglyceride (28.7%) concentrations (Figure 5D–G). Histological analysis showed a significant reduction in the development of fatty liver (Figure 5H). All these changes occurred even though dietary intake remained the same.

The gut microbiota analysis showed that consumption of the HFS + FF diet increased α-diversity with respect to the HFS group, indicating that the presence of omega-3 fatty acids, antioxidants, plant proteins, polyphenols, soluble and insoluble fiber had a beneficial effect on the gut microbiota (Figure 6A). We demonstrated that consumption of HFS diet increased the abundance of *Blautia producta* and *Ruminococcus flavefaciens*, while the addition of functional foods to the HFS diet significantly increased the abundance of other species, in particular *Prevotella copri*, *Akkermansia muciniphila*, *Ruminococcus bromii*, *Faecalibacterium prausnitzii* and *Mucispirillum schaedleri* which are involved in complex carbohydrate degradation, and are associated with improved insulin sensitivity, anti-inflammatory response and improved epithelial barrier integrity (Figure 6B). In fact, the addition of functional foods to the HFS diet significantly decreased pro-inflammatory biomarkers and increased the abundance of occludins in the colon (Figure 6C,D). To assess inflammation in the colon that might be involved in the development of metabolic endotoxemia, we measured intestinal CD14, as this is essential for the recognition of LPS by the TLR4 complex and the subsequent generation of systemic inflammation. The HFS group showed a significant increase in intestinal CD14 abundance; however, the addition of functional foods decreased the abundance of this biomarker, and it was associated with a decrease in serum LPS (Figure 6C–E). As expected, the addition of functional foods (FF) to the HFS diet significantly decreased the concentration of hepatic 2-OG (Figure 6F). As a consequence, the addition of FF to the diet reduced the abundance of proteins involved in lipogenesis (Figure 6G,H), as well as hepatic triglyceride and cholesterol concentrations, reducing the macroscopic accumulation of hepatic lipids (Figure 6I,J). Interestingly, the addition of FF modified the hepatic lipid profile of fatty acids, particularly an increase in eicosapentaenoic acid (EPA) (Figure 6K).

### 3.6. The Combination of Bioactive Compounds in Functional Foods Stimulated Hepatic Fatty Acid Oxidation and Improved Mitochondrial Function

Since there is strong evidence that mitochondrial dysfunction plays a significant role in the development of MASLD, we were interested in evaluating the effects of bioactive compounds present in the experimental diet on mitochondrial oxidative function. The results showed that the consumption of functional foods increased the abundance of carnitine palmitoyl transferase-1 (CPT-1), a key enzyme involved in the mitochondrial oxidation of fatty acids (Figure 7A). In fact, isolated hepatocytes incubated with an ethanolic extract of the functional foods showed a dose-response increase in oxygen consumption rate (OCR), particularly maximal respiration and mitochondrial spare respiratory capacity (Figure 7B,C). In addition, consumption of functional foods decreased liver reactive oxygen species (ROS) by 59% despite consumption of the HFS diet (Figure 7D), and this was accompanied by a significant increase in the protein abundance of the transcription factor Nrf2 and the antioxidant enzymes SOD2, catalase, and GPx4, involved in the scavenging of ROS from mitochondria (Figure 7E,F). This evidence suggests that the consumption of functional foods protects the liver from oxidative stress and lipid accumulation.

### 3.7. MASLD Increases Blautia Producta and 2-Oleoyl Glycerol in Humans

To investigate whether the gut microbiota was altered in subjects with MASLD, we studied 10 subjects with the following characteristics, shown in Table 2. The results showed that gut microbiota α-diversity was significantly lower in subjects with MASLD than in the control group (Figure 8A). The PCoA indicated that the gut microbiota of the subjects with MASLD was dissimilar to that of the healthy subjects (Figure 8B). When the taxonomic assignment was measured at the genus level, we found a significant increase in the genus Blautia, Eubacterium, Subdoligranulum, Fusicatenibacter, Agathobacter and Allistipes (Figure 8C). Interestingly, Blautia was the genus with the highest abundance, 37.5-fold higher than in the healthy subjects. LDA showed that the most significantly abundant bacterium at the species level was *Blautia producta* (Figure 8D). As in the rat study, Blautia producta was the most abundant species in subjects with MASLD, which was associated with the production of the metabolite 2-OG. As shown in (Figure 8E), the serum concentration of 2-OG was 10.5-fold higher in subjects with MASLD than in healthy subjects. These results suggest that 2-OG could be used as a biomarker for the presence of hepatic steatosis. As shown in (Figure 8F), there was a significant correlation between *B. producta* and 2-OG and between 2-OG and *B. producta* with body mass index.

## 4. Discussion

MASLD has emerged as the most common liver disease worldwide as obesity-related disorders and type 2 diabetes are on the rise [20]. Liver disease accounts for two million deaths annually and is responsible for 1 in 25 deaths worldwide [21]. It is known that diet plays an important role in lipid accumulation in the liver. Diets rich in fats and sugars lead to an increased lipid synthesis modulated mainly by insulin [22], which can be magnified by the pro-inflammatory state that occurs in the liver [23]. Also, dysbiosis of the gut microbiota has been associated with the development of MASLD [24,25,26]. Evidence has shown that dysbiosis of the gut microbiota during obesity and insulin resistance produced metabolites associated with lipid accumulation in the liver [27]. It was recently demonstrated that the pathogenesis of non-alcoholic steatohepatitis was associated with the production of 2-OG, a metabolite produced by *Blautia producta*, which is associated with the promotion of liver inflammation and liver fibrosis through the activation of macrophages and subsequent activation of hepatic stellate cells [19]. The present study demonstrated a significant increase in *Blautia producta* in rats fed the HFS diet or in subjects with MASLD with a significant increase in liver and serum 2-OG. As a new finding, this study showed that 2-OG has a new metabolic function in hepatocytes by activating the abundance of lipogenic enzymes and decreasing mitochondrial activity, indicating that 2-OG is a molecule that stimulates the development of fatty liver by increasing lipid accumulation. Interestingly, incubation of hepatocytes with 2-OG decreased mitochondrial function, which has been associated with various diseases [28]. Therefore, new strategies for the management of fatty liver disease could consider the measurement of 2-OG concentration as a biomarker. As observed in the present study, animals fed the HFS diet produced hepatic macrovesicular steatosis, together with a slight decrease in hepatic CPT-1. It was also shown that hepatic reactive oxygen species (ROS) increased significantly by 300%, indicative of mitochondrial dysfunction in which electrons are incompletely transferred to complex I or III of the electron transport chain in the mitochondria, and ROS are generated. As a consequence, it was observed that after consumption of the HFS diet, there was an increase in the abundance of pro-inflammatory cytokines, in particular IL-1β, TNFα, and IL-6, which has been reported previously [29].

The recommended therapies for MASLD are lifestyle changes [30]. Due to their properties, the use of plant-derived compounds in the prevention and treatment of liver diseases has attracted considerable attention for their effects on the gut microbiota [31]. The gut-liver interaction has been recognized as a very important pathway in the regulation of liver function. Therefore, a decrease in the abundance of *Blautia producta* in the intestinal microbiota and a consequent decrease in 2-OG production should be considered in nutritional strategies to attenuate the progression of fatty liver. Interestingly, the addition to the diet of functional foods with plant proteins, rich in antioxidants and linolenic acid, significantly decreased the concentrations of pro-inflammatory cytokines.

Furthermore, the addition of functional foods to the HFS diet increased mitochondrial activity and activated the transcription factor Nfr2 and the antioxidant enzymes SOD2 and GPx4 to ensure neutralization of ROS. In the present study, 2-OG, a metabolite produced by gut microbiota, was shown to decrease mitochondrial oxidative capacity in hepatocytes; however, an extract of the functional foods reactivated mitochondrial oxidative capacity.

Our study showed an additional adverse effect of 2-OG that stimulated de novo lipogenesis since in cultured hepatocytes, as well as in the experimental rat model of hepatic steatosis, 2-OG was shown to increase ChREBP and SREBP1 concentrations, which are transcription factors that regulate the expression of lipogenic genes [32]. In fact, our results in vitro and in vivo showed that 2-OG increased FAS, SCD1, and AGPAT6 after the consumption of an HFS diet and decreased pACC, which is the active form of ACC that increases fatty acid synthesis. Interestingly, with the HFS + FF diet, these effects were reversed and associated in part with a decrease in *Blautia producta* and 2-OG. We have evidence that some of the functional foods included in the present study, such as nopal [12] or soy protein [9], decrease the abundance of *B. producta*, indicating that the combination of functional foods is responsible for decreasing this species. However, it cannot be ruled out that the presence of some other compounds in functional foods may have a synergistic effect, such as the α-linolenic acid present in chia seed, which also suppresses the expression of the ChREBP gene and/or a defect in its nuclear translocation [33] and prevents hepatic steatosis [34]. Alpha-linolenic acid present in chia oil is the dietary precursor of the long-chain omega-3 PUFAs EPA and DHA [10,35]. Also, curcumin has been reported to reduce total triglycerides and waist circumference in NAFLD [36]. Surprisingly, there was a significant increase in hepatic DHA and EPA following functional food consumption, suggesting that the presence of long-chain fatty acids may contribute in part to decreased hepatic de novo lipogenesis, inflammation, and oxidative stress and a significant increase in β-oxidation, as previously described [37,38]. Decreased hepatic lipogenesis has been reported to have a beneficial effect against MASLD [38]. Since there is no pharmacological treatment for MASLD [39], the development of this functional food combination that eliminates *Blautia producta* from the gut microbiota and decreases 2-OG concentration, which also increases mitochondrial function as well as the abundance of *Faecalibacterium prausnitzii* associated with a reduced inflammation could be a dietary strategy to be considered in future studies, since several drugs induce mitochondrial dysfunction and may exacerbate MASLD in some patients [40].

## 5. Conclusions

The present study showed that there is a significant increase in 2-OG in patients with MASLD, associated with a dysbiosis of the gut microbiota where *Blautia producta* increased significantly. It suggests that 2-OG determination could be used as a predictor for the development of fatty liver, and future studies are required to demonstrate if the 2-OG concentrations are proportional to the degrees of hepatic steatosis. Further studies are necessary to demonstrate that dietary strategies with functional foods are effective in subjects with MASLD.

## Figures and Tables

**Figure 1 nutrients-16-02594-f001:**
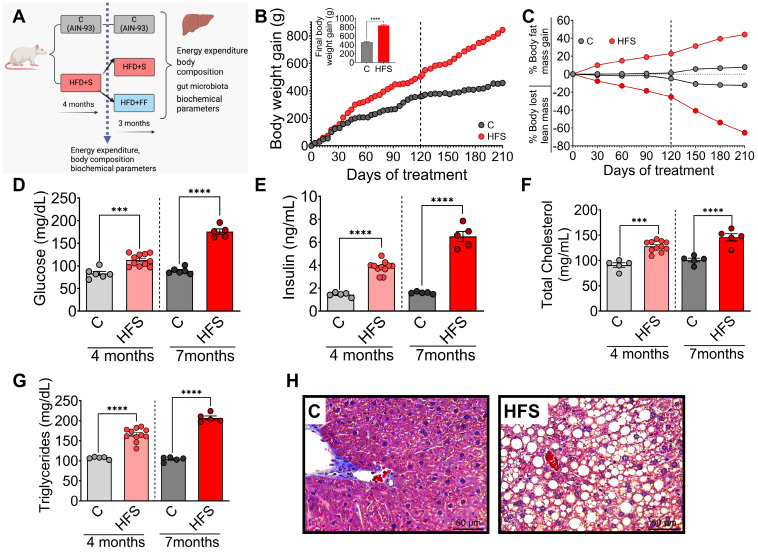
Consumption of a high-fat diet +5% sucrose in the drinking water (HFS) produces hepatic steatosis. (**A**) Experimental model of hepatic steatosis, (**B**) Body weight gain, (**C**) Body composition, serum (**D**) Glucose, (**E**) Insulin, (**F**) Total cholesterol, (**G**) Triglycerides, and (**H**) Histological analysis of liver from rats fed HFS or Control diet (C) for 7 months. Mean ± SEM is shown in each graph, n = 6–7 in each group. Significant differences are presented by asterisk, *** *p* < 0.0002, **** *p* < 0.00001.

**Figure 2 nutrients-16-02594-f002:**
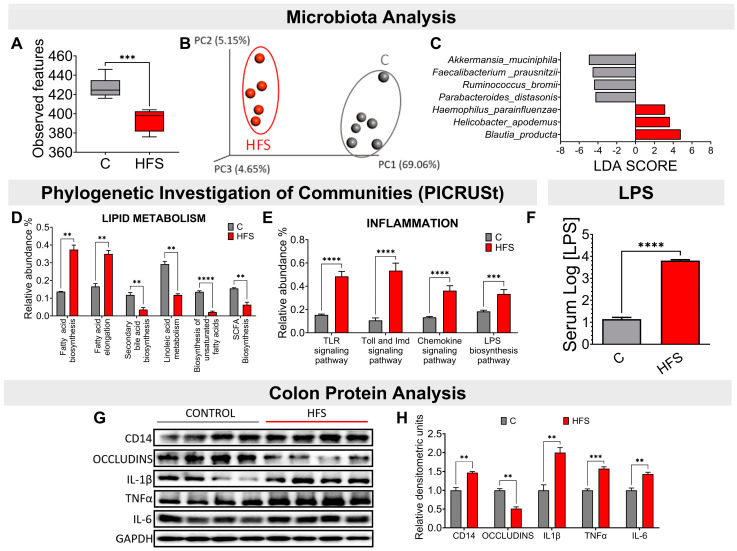
Consumption of a high-fat +5% sucrose diet in the drinking water (HFS) promotes dysbiosis of the gut microbiota and a chronic inflammatory state. (**A**) Alpha diversity, (**B**) Principal Component Analysis, (**C**) Linear Discriminant Analysis, Prediction of metagenome functionality (PICRUST) of (**D**) lipid metabolism and (**E**) inflammation. (**F**) serum LPS, (**G**) Western blot, and (**H**) Densitometric analysis of colonic inflammatory proteins extracted from rats fed HFS or C diet for 7 months. Mean ± SEM is shown in each graph, n = 6–7 in each group. Significant differences are presented by asterisk, ** *p* < 0.0021, *** *p* < 0.0002, **** *p* < 0.00001.

**Figure 3 nutrients-16-02594-f003:**
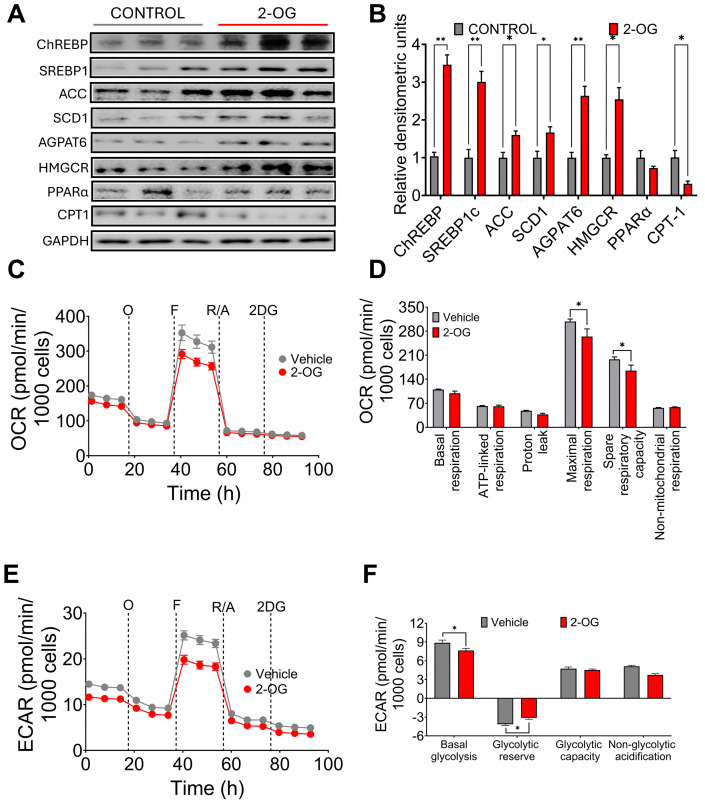
Effect of 2-oleoyl glycerol (2-OG) on rat hepatocytes. (**A**) Western blot and (**B**) Densitometric analysis of hepatic lipogenic proteins, (**C**) Oxygen consumption rate, (**D**) Mitochondrial function parameters, (**E**) Extracellular acidification rate, and (**F**) Cellular glycolysis analysis in hepatocytes cultured with vehicle or 2-OG. Mean ± SEM is shown in each graph. Significant differences are presented by asterisk, * *p* < 0.0332, ** *p* < 0.0021.

**Figure 4 nutrients-16-02594-f004:**
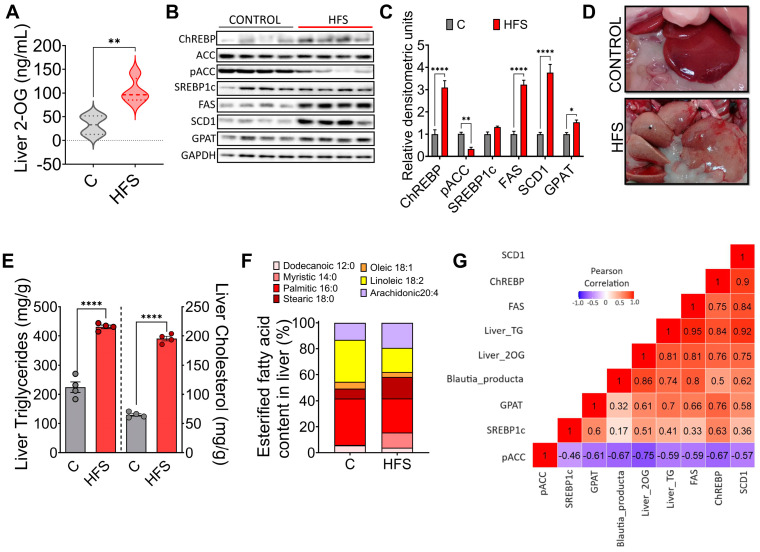
2-oleoyl glycerol (2-OG) stimulates lipogenesis in rats fed high fat +5% sucrose in the drinking water (HFS). (**A**) hepatic 2-OG concentration, (**B**) Western blot analysis, (**C**) Densitometric analysis of transcription factors and target enzymes of lipogenesis, (**D**) Macroscopic view of liver, (**E**) Hepatic triglycerides and cholesterol concentrations, (**F**) Hepatic lipid profile and (**G**) Analysis of correlations of lipogenic proteins, 2-OG concentration and *Blautia producta*, in rats fed a control diet or HFS diet. Mean ± SEM is shown in each graph. Significant differences are presented by asterisk. * *p* < 0.0332, ** *p* < 0.0021, **** *p* < 0.0001.

**Figure 5 nutrients-16-02594-f005:**
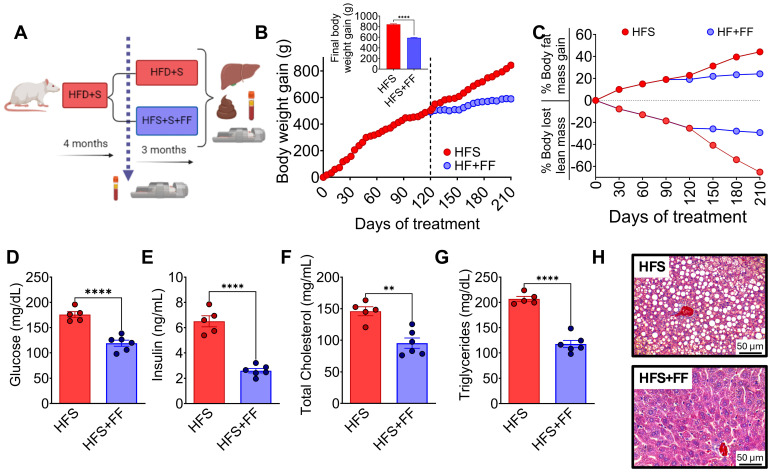
Effect of a combination of functional foods on hepatic steatosis. (**A**) Experimental model, (**B**) Body weight gain, (**C**) Body composition, serum fasting, (**D**) Glucose, (**E**) Insulin, (**F**) Total cholesterol and (**G**) triglycerides, and (**H**) histological analysis of liver from rats fed HFS diet with or without functional foods for 3 months. Mean ± SEM is shown in each graph. Significant differences are presented by asterisk, ** *p* < 0.0021, **** *p* < 0.0001.

**Figure 6 nutrients-16-02594-f006:**
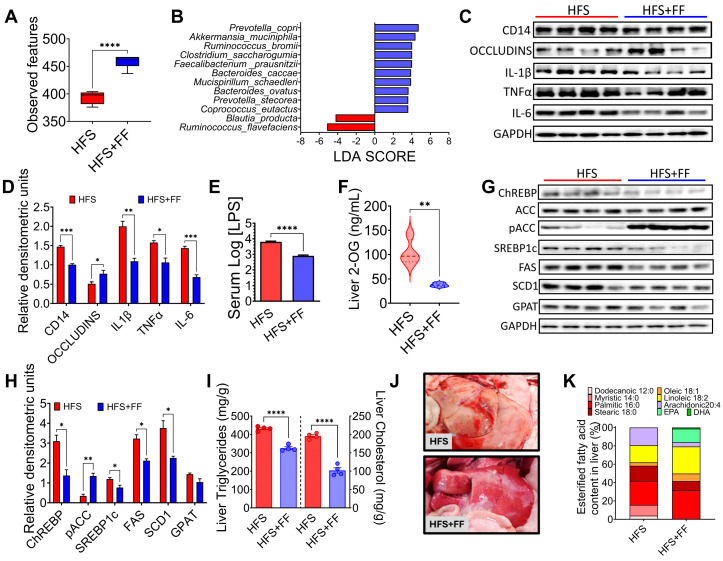
Functional foods modify gut microbiota, attenuating hepatic steatosis. (**A**) Alpha diversity, (**B**) Linear Discriminant Analysis, (**C**) Western Blot and (**D**) Densitometric analysis of hepatic pro-inflammatory cytokines, (**E**) Serum LPS concentration, (**F**) Hepatic concentration of 2-oleoyl glycerol (2-OG), (**G**) Western blot and (**H**) Densitometric analysis of hepatic lipogenic proteins. (**I**) Macroscopic view of the liver, (**J**) hepatic triglycerides and cholesterol, and (**K**) Hepatic lipid profile of rats fed HFS with or without functional foods for 3 months. Mean ± SEM is shown in each graph. Significant differences are presented by asterisk, * *p* < 0.0332, ** *p* < 0.0021, *** *p* < 0.0002, **** *p* < 0.0001.

**Figure 7 nutrients-16-02594-f007:**
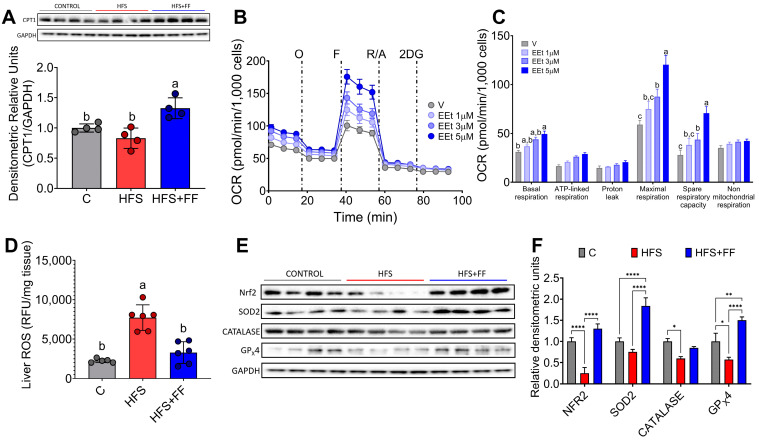
Consumption of Functional foods in the diet improves mitochondrial function. (**A**) Hepatic CPT-1 protein abundance, (**B**) Oxygen consumption rate and (**C**) Mitochondrial function parameters in hepatocytes incubated with different concentrations of the functional food extract, (**D**) Reactive oxygen species, (**E**) Western blot and (**F**) Densitometric analysis of the transcription factor Nrf 2 and antioxidant enzymes SOD2, catalase, GPx4 in liver of rats fed HFS diet with or without functional foods for 3 months. Mean ± SEM is shown in each graph. Significant differences are presented by asterisks * *p* < 0.0332, ** *p* < 0.0021, **** *p* < 0.0001, or letters (a > b > c).

**Figure 8 nutrients-16-02594-f008:**
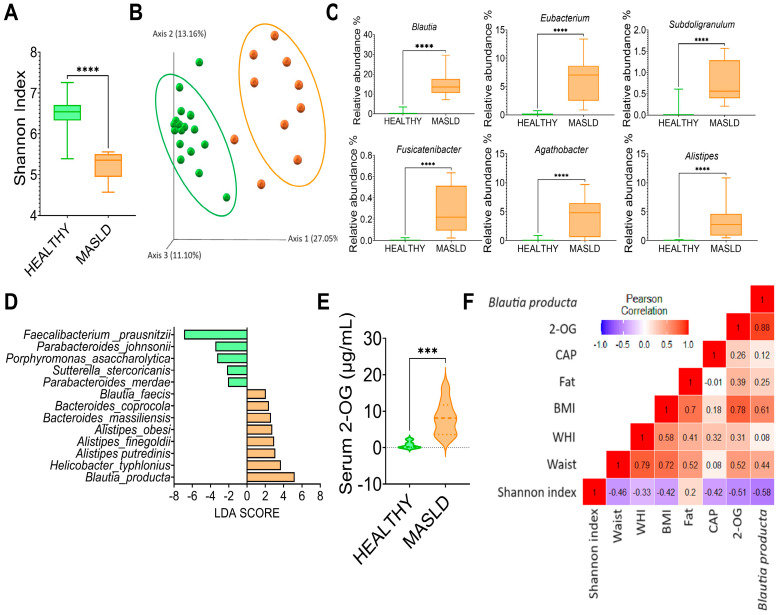
Subjects with MASLD increase *Blautia producta* and serum concentrations of 2-OG. (**A**) Alpha diversity, (**B**) Principal component analysis, (**C**) Most abundant genus in subject with MASLD, (**D**) Species Linear discriminant analysis between healthy and MASLD subjects, (**E**) Serum 2-OG concentration, and (**F**) correlation analysis between anthropometric variables, elastography parameters, 2-OG, and *Blautia producta* in subjects with MASLD. Mean ± SEM is shown in each graph. Significant differences are presented by asterisk, *** *p* < 0.0002, **** *p* < 0.0001.

**Table 1 nutrients-16-02594-t001:** Experimental diet composition.

Ingredients (%)	C	HFS	HFS + FF
Cornstarch	39.775	23.903	38.7
Casein	20.000	24.000	24.000
Dextrinized cornstarch	13.200	10.267	10.267
Sucrose	10.000	7.778	7.778
Soybean oil	10.000	7.000	7.000
Fiber Celluose	5.000	5.000	-
Mineral Mix AIN-93MX	3.500	3.500	3.500
Vitamin Mix AIN-93-VX	1.000	1.000	1.000
L-Cystine	0.300	0.300	0.300
Choline bitartrate	0.2500	0.2500	0.2500
Tert-butylhydroquinone	0.0014	0.0014	0.0014
Lard	-	17	17
Chia oil	-	-	3
Nopal	-	-	5
Soy protein	-	-	20
Curcumin	-	-	1

**Table 2 nutrients-16-02594-t002:** Characteristics of the subjects.

Group	Healthy	MALFD
Age (years)	33.9 ± 3.5	49.9 ± 3.4
Sex% (Female/Male)	80/20	80/20
Weight (kg)	66.2 ± 3.7	96.3 ± 3
BMI (kg/m^2^)	21.9 ± 1.28	39.3 ± 1.9
Body fat (%)	24.8 ± 1.7	40.1 ± 1.9
Glucose (mg/dL)	83.5 ± 5.5	124.4 ± 13.3
Triglycerides (mg/dL)	114.3 ± 7.6	188.2 ± 26
Total cholesterol (mg/dL)	155.9 ± 5.2	194.5 ± 10.2
AST (IU/L)	22.8 ± 2.4	33.3 ± 5
ALT(IU/L)	25.7 ± 3.2	44.1 ± 9
CAP score (dB/m)	<294	355.1 ± 6.3
Fibroscan (kPa)	<8.2	35 ± 1.2

## Data Availability

The 16S rRNA gene sequencing raw sequence reads (FASTQ) are available at the NCBI Sequence Read Archive with the BioProject ID PRJNA1127599 and PRJNA1127639 (https://dataview.ncbi.nlm.nih.gov/object/PRJNA1127599?reviewer=ulv6r50fgo1umq61ri1ipkgar2, accessed on 4 August 2024)

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
