# Peer review of "Hepatic Steatosis Can Be Partly Generated by the Gut Microbiota–Mitochondria Axis via 2-Oleoyl Glycerol and Reversed by a Combination of Soy Protein, Chia Oil, Curcumin and Nopal"

_nutrients, 2024, doi:10.3390/nu16162594_

Round 1

Reviewer 1 Report

Comments and Suggestions for Authors

A well-prepared scientific work presenting the role of 2-oleolyl glycerol in the development of hepatic steatosis.   The objective of the study was to check how high-fat sucrose diet (HFS) affect the microbiome lending to microbiological balance disturbance increase level of 2-oleolyl glycerol and in consequence liver steatosis.   Studies on rodent animal models have been enriched with in vitro analysis and examinations performed on humans.   Themanuscript brings a new knowledge about the role of 2-oleolyl glycerol in liver steatosis formation, especially changes in mitochondria and functioning and energy metabolism.   Authors achieved their goal by showing the mechanism by which 2-oleolyl glycerol leads to liver steatosis.   This phenomenon is based on lipid accumulation caused by activation of lipogenic enzymes and reduced mitochondrial activity.   What is interesting authors show also that functional food additives contain mainly chia oil, nopal, soy protein and curcumin eliminates B. producta from the gut microbiota, decreases 2-oleolyl glycerol level, increase mitochondrial function and the abundance of F. prausnitzii leading to inhibition of accumulation of lipids in the liver.

Minor comments

- not good quality of figures, must be improved

- figure 5H is incomplete, also figure 6I is uninformative if there is no comparison with HFS and preferably also control group

- check carfully editorial mistakes, double spacing etc.

Author Response

  1. Figures were changed to the JPG format with 1200 dpi of resolution
  2. We include a histological image of the HFS group in figure 5H and macroscopial view of the liver in the HFS group.

We double-check the editorial mistakes

Reviewer 2 Report

Comments and Suggestions for Authors

According to the manuscript titled "Hepatic steatosis is generated by the gut microbiota-mitochondria axis via 2-oleolyl glycerol and reverted by functional foods" by Mónica Sánchez-Tapia and colleagues. The metabolic dysfunction-associated steatotic liver disease (MASLD) is a serious health issue, and recent research indicates that gut microbiota play an important role in its development. Hepatic fibrosis is associated with 2-oleoyl glycerol (2-OG) produced by the gut microbiota, but it is not known whether this metabolite contributes to hepatic steatosis. It was the aim of this study to determine whether a high-fat sucrose diet (HFS) increases 2-OG production through dysbiosis of the gut microbiota and whether this metabolite modulates hepatic lipogenesis and mitochondrial activity for the development of hepatic steatosis as well as whether functional foods can reverse this process. For seven months, Wistar rats were fed a HFS diet. We evaluated the effects of HFS with functional foods on body composition, biochemical parameters, gut microbiota, lipogenic and antioxidant enzyme protein abundance, liver 2-OG, and mitochondrial function in rats. The gut microbiota and serum 2-OG were analyzed in humans with MASLD. A HFS diet administered to Wistar rats caused oxidative stress, hepatic steatosis and dysbiosis of the gut microbiota, decreasing the microbial diversity and increasing the abundance of Blautia producta, which in turn increased 2-OG levels. Through ChREBP and SREBP-1, this metabolite increased de novo lipogenesis. A significant increase in mitochondrial dysfunction was observed with 2-OG. With the addition of functional foods to the diet, the gut microbiota was modified, resulting in a reduction in B product levels and 2-OG levels, which resulted in a decrease in body weight gain, body fat mass, serum glucose, insulin, cholesterol, triglycerides, as well as an increase in mitochondrial function. In order to use 2-OG as a biomarker, this metabolite was measured in healthy subjects or with MASLD and it was demonstrated that subjects with hepatic steatosis II and III had significantly higher 2-OG than healthy subjects, suggesting that the abundance of this circulating metabolite could be an indicator of steatosis in the liver. I would like to make a few comments regarding the present manuscript.

  • It is possible that the introduction summarizes too much. The main outcome of the study should be described in more detail. The title might be changed to another no direct option with might or should.

  • In the animal study, how was the sample size calculated?

  • A total of 24 healthy subjects and ten patients with MASLD will be enrolled in the study. With MASLD being the focus of the study, the number of subjects should be higher in this group, is that possible?

  • Could the author add more detailed information about the ELISA kit numbers or reference, the concentration of hematoxylin-eosin in the biochemical parameters and the histological procedures?

  • The authors provide general information about the bioinformatic analysis procedure. Sequence quality, programs used, and QIIME2 scripts are examples of missing information

  • The resolution of Figure 1 should be improved, especially in Figure 1A.

  • It is noted that the authors have mentioned PICRUSt analyses, but no information is provided in the material and methods section. Please provide this information.

  • Again, there is a problem with the figure resolution. The authors were able to utilize the entire page with the figure.It is not a problem, but Figure 2 is difficult to understand

  • The main issue with the present manuscript is the resolution of the figures. There is a reason why the blots have the colors they do, for example, see figures 2, 4, 6, and 7. The colors red and yellow and others.

  • Please ensure that italics are used when naming species or any microbial entity.

  • There may be a need to include Table S1 in the main manuscript.

  • In the supplemental information, there is no information on how functional food was planned. Maybe this information is crucial to the paragraph in the introduction.

  • To ensure that the size is correctly assessed, the blots should be uploaded uncut and in a single file.

  • Based on Figure 6E, there were no differences in LPS levels; do the authors have any explanation for this?

  • It is unclear how the authors could attribute all the microbial changes to a single bacteria if they used only a fragment of 600 bp (V3-V4, 16S rRNA sequencing), which analysis is the basis of their argument

Author Response

Reviewer 2

1. The introduction section was modified as suggested by the reviewer

2. The sample size was calculated according to the next formula:

? =

2?&(?? + ??)&

Δ&

In which, n is the value of the sample size, s corresponds to the value of the standard

deviation, Zα (Z for alpha) refers to the type I error (confidence level α= 0.05

corresponding to a value of Z=1.96), Zβ (Z for beta) with a power of 80% (value of

Z=0.84) and Δ to the difference in magnitude between treatment means (amplitude). In

the methods section was included formula for comparison of means was used

3. Thank you for the suggestion. Only 10 subjects with MASLD were included in this

preliminary study to confirm that 2-oleyl glycerol was significantly high with respect to

the healthy subjects. Future studies in subjects with different states of hepatic steatosis

are needed to establish values for each different state of hepatic steatosis.

4. The numbers of the kits were included in the method section as well as the percentage

hematoxylin-eosin in the histological analysis.

5. The QIIME 2 scripts were:

conda activate qiime2-2022.2

qiime tools import \

--type 'SampleData[PairedEndSequencesWithQuality]' \

--input-path data/read_manifest.csv \

--output-path artifacts/Reads.qza \

--input-format PairedEndFastqManifestPhred33

qiime demux summarize \

--i-data artifacts/Reads.qza \

--o-visualization artifacts/ReadQC.qzv

qiime dada2 denoise-paired \

--i-demultiplexed-seqs artifacts/Reads.qza \

--p-trunc-len-f 288 \

--p-trunc-len-r 270 \

--p-trim-left-f 10 \

--p-trim-left-r 10 \

--p-n-threads 8 \

--o-table artifacts/Feature_Table.qza \

--o-representative-sequences artifacts/Feature_Sequences.qza \

--o-denoising-stats artifacts/Feature_Statistics.qza \

--verbose

qiime metadata tabulate\

--m-input-file artifacts/Feature_Statistics.qza \

--o-visualization denoising-stats.qzv

qiime feature-table summarize \

--i-table artifacts/Feature_Table.qza \

--o-visualization table.qzv \

--m-sample-metadata-file data/sample-metadata.tsv

qiime feature-table tabulate-seqs \

--i-data artifacts/Feature_Sequences.qza \

--o-visualization rep-seqs.qzv

qiime phylogeny align-to-tree-mafft-fasttree \

--i-sequences artifacts/Feature_Sequences.qza \

--o-alignment aligned-rep-seqs.qza \

--o-masked-alignment masked-aligned-rep-seqs.qza \

--o-tree unrooted-tree.qza \

--o-rooted-tree rooted-tree.qza

qiime diversity core-metrics-phylogenetic \

--i-phylogeny rooted-tree.qza \

--i-table artifacts/Feature_Table.qza \

--p-sampling-depth 40214 \

--m-metadata-file data/sample-metadata.tsv \

--output-dir core-metrics-results

qiime diversity alpha-group-significance \

--i-alpha-diversity core-metrics-results/faith_pd_vector.qza \

--m-metadata-file data/sample-metadata.tsv \

--o-visualization core-metrics-results/faith-pd-group-significance.qzv

qiime diversity alpha-group-significance \

--i-alpha-diversity core-metrics-results/shannon_vector.qza \

--m-metadata-file data/sample-metadata.tsv \

--o-visualization core-metrics-results/shannon-group-significance.qzv

qiime diversity beta-group-significance \

--i-distance-matrix core-metrics-results/bray_curtis_distance_matrix.qza \

--m-metadata-file data/sample-metadata.tsv \

--m-metadata-column Description \

--o-visualization core-metrics-results/bray2-significance.qzv \

--p-pairwise

qiime diversity beta-group-significance \

--i-distance-matrix core-metrics-results/unweighted_unifrac_distance_matrix.qza \

--m-metadata-file data/sample-metadata.tsv \

--m-metadata-column Description \

--o-visualization core-metrics-results/weighted-unifrac-significance.qzv \

--p-pairwise

qiime diversity alpha-rarefaction \

--i-table artifacts/Feature_Table.qza \

--i-phylogeny rooted-tree.qza \

--p-max-depth 40214\

--m-metadata-file data/sample-metadata.tsv \

--o-visualization alpha-rarefaction.qzv

qiime feature-classifier classify-sklearn \

--i-classifier classifier/silva-138-99-nb-classifier.qza \

--i-reads artifacts/Feature_Sequences.qza \

--o-classification taxonomy_silva.qza

qiime metadata tabulate \

--m-input-file taxonomy_silva.qza \

--o-visualization taxonomy_silva.qzv

qiime taxa barplot \

--i-table artifacts/Feature_Table.qza \

--i-taxonomy taxonomy_silva.qza \

--m-metadata-file data/sample-metadata.tsv \

--o-visualization taxa-bar-plots-silva.qzv

With this analysis we can have quality plots, assigned sequences, phylogenetic tree,

different metrics of alpha and beta diversity and taxonomy assignments by SILVA.

It is difficult to include this information in the manuscript.

6. The resolution of all figures was improved and the format was changed to JPEG.

7. PICRUSt analysis information was added in the material and methods section.

8. Figure 2 resolution was improved and enlarged suggested by the reviewer.

9. The format and resolution of all figures were changed to JPG. In addition, there was no

reason for the colors in the blots. Blots were modified to have a single color (black).

10. The manuscript was checked for naming species with italics.

11. Table S1 is now included in the main manuscript as Table 1.

12. Thank you for the suggestion. We include in the introduction section a paragraph

providing information about the functional foods that we used in the study

13. The blots were uploaded, uncut in a single file called “blots” when we submitted the

manuscript.

14. As you can see in Fig 6E, the group fed HFS+Functional food showed a significant

decrease in LPS compared to those fed the HFS diet (from approximately 6000 ng to

800ng), for this reason, the Y axis was transformed to log units.

15. Previous studies have demonstrated that bacterial 16S rRNA gene amplicons were

constructed via amplification of just a one region, the V4 region of 16S rRNA gene and

they found also an increase in Blautia producta in non-alcoholic steatohepatitis (nature

communications. https://doi.org/10.1038/s41467-023-35861-1), similar to our results

Rarefaction analyses indicate an adequate depth of readings (minimum 40214) per

sample.

Round 2

Reviewer 2 Report

Comments and Suggestions for Authors

I would like to thank the authors for providing information during the revision process. There may be differences in the form in which the response is provided across countries. Point-by-point responses are an effective method of organizing and understanding reviewer questions, and the information may or may not be included in the main manuscript. This process cannot be improved by providing simple responses and not including the information in the main manuscript. 

For example, as a result of the review process, the authors did not use references to the materials and methods in explaining the results (PICRUSt). Maybe the authors should consider adding additional information raised in the questions to the main document

Round 3

Reviewer 2 Report

Comments and Suggestions for Authors

Thank you to the authors for taking the time and effort to respond to the reviewer's queries. I have no further comments to make